# Significant Factors for Modelling Survival of *Escherichia coli* in Lake Sediments

**DOI:** 10.3390/microorganisms12061192

**Published:** 2024-06-13

**Authors:** Ichiro Yoneda, Masateru Nishiyama, Toru Watanabe

**Affiliations:** 1Department of Regional Environment Creation, United Graduate School of Agricultural Sciences, Iwate University, 18-8 Ueda 3-Chome, Morioka 020-8850, Japan; u3321010@iwate-u.ac.jp; 2Department of Food, Life and Environmental Sciences, Faculty of Agriculture, Yamagata University, 1-23 Wakaba-Machi, Tsuruoka 997-8555, Japan; m-nishiyama@tds1.tr.yamagata-u.ac.jp

**Keywords:** *Escherichia coli*, population change rate, fecal indicator bacteria, numerical model, lake sediment, health risk

## Abstract

Currently available numerical models that describe the fecal contamination of aquatic environments using *Escherichia coli* as an indicator bacterium did not consider its survival in sediments. We conducted a series of comparative experiments to reveal the independent and interactive effects of sediment factors, including temperature, pH, water-extractable total dissolved solids (TDSs), coexisting microbes, and sampling sites, in lake environments on *E. coli* survival. In experiments, *E. coli* survival was observed by controlling any two factors at a time. Consequently, the decrease in pH and presence of coexisting microbes enhanced *E. coli* die-off, whereas the addition of water-extractable TDSs promoted its growth. To select factors to be considered for modelling *E. coli* survival in sediments, the independent effects of each factor and the interaction effect of the two factors were statistically compared based on their effect sizes (η^2^). As a result, pH (η^2^ = 59.5–89.0%) affected *E. coli* survival most significantly, followed by coexisting microbes (1.7–48.4%). Among the interactions affecting *E. coli* survival, including pH or coexisting microbes—which had larger independent effects—relatively larger statistically significant interactions were observed between pH and coexisting microbes (31.1%), coexisting microbes and water-extractable TDSs (85.4%), and coexisting microbes and temperature (26.4%).

## 1. Introduction

Contamination of aquatic environments such as lakes, rivers, and coastal areas with fecal pathogens can be caused by diverse sources, such as the discharge of treated sewage from wastewater treatment plants, surface runoff from urban and agricultural areas, and wildlife feces [1,2,3]. In aquatic environments, fecal pathogens derived from these sources can attach to sediment particles, which are then deposited and accumulate on the channel bed [4], thus making sediments a potential reservoir of fecal pathogens. In addition, contaminated sediments become a source of these pathogens when resuspended by floods or artificial disturbance [5,6]. Understanding the survival of fecal pathogens in sediments and water is important because water contaminated with fecal pathogens poses a risk to human health when used for drinking, recreation, and irrigation.

*Escherichia coli* is a widely used indicator of fecal contamination in aquatic environments [7,8] because its detection indicates the possible presence of pathogens originating from human and animal feces. However, although *E. coli* in sediments can contaminate surface water, this fact is not considered in evaluating the quality of aquatic environments. In addition, while most *E. coli* strains are harmless, some strains, such as diarrheagenic *E. coli*, can cause human diseases [9,10]. Among them, antibiotic-resistant diarrheagenic *E. coli* threatens human health, as those resistant to third-generation cephalosporin and carbapenem are classified as critical-priority bacteria by the World Health Organization (WHO) [11]. Previous studies have reported that bacteria, including *E. coli*, can survive for long periods and even grow in sediments associated with or by forming biofilms and that biofilms could serve as a hot spot for bacteria to obtain antibiotic-resistance genes [12]. These reports imply that sediments in aquatic environments play a role as a potential reservoir of antibiotic-resistant and/or diarrheagenic *E. coli* as well as harmless ones.

The survival of *E. coli* in sediments is affected by various factors, including temperature, pH, organic carbon content, predation by protozoa, competition with other bacteria, and sediment texture. Previous studies have reported that *E. coli* can survive in sediments for a long period at low temperatures [13,14], neutral pH [15,16], high organic carbon content [17,18], and small particle size [14,19], whereas predation and competition [20,21] shorten their survival in sediments. These previous studies include, for example, Garzio-Hadzick [14] assessing the effect of temperature (4, 14, and 24 °C) on *E. coli* survival, Chandran et al. [17] evaluating its survival in three different types of sediments with different particle sizes and organic carbon content at 25 °C, Smith et al. [19] reporting the effect of temperature oscillations from 17 °C to 28 °C and sediment texture between loamy sand and clay loam, and Wanjugi et al. [21] evaluating the effect of predation and competition on its survival under conditions at 12 and 30 °C. Although the independent effects of sediment factors on the survival of *E. coli* have been investigated in previous studies, they have never been compared, and their interactive effects remain unclear.

Water in artificial and natural waterbodies, including lakes, ponds, and dam reservoirs, is often used for irrigation and drinking [22], and its contamination with *E. coli*, including antibiotic-resistant and/or diarrheagenic strains, can pose risks to human health. However, sediments in these waterbodies can provide long-term survival conditions for *E. coli* (e.g., high organic carbon content and small particle size), and their disturbance can cause *E. coli* contamination of lake water source used for irrigation and drinking. Numerical models simulating the survival of such *E. coli* in aquatic environments have been developed and used to predict water contamination by them [23,24,25,26]. These models often consider the effect of factors, including temperature and light intensity, on *E. coli* survival in water and sedimentation/resuspension of *E. coli*. However, these models do not consider the effect of sediment factors on the survival of *E. coli* in the sediments. For more reliable simulation, the sediment factors that strongly affect the survival of *E. coli* should be determined and then should be considered in the simulation.

This study aimed to evaluate the independent and interactive effects of sediment factors on *E. coli* survival, to select factors that strongly affect its survival in a lake environment, and to feed numerical models that simulate its survival with information for their developments. We conducted laboratory experiments to observe *E. coli* survival in sediments under conditions that controlled two of the following factors at a time: temperature, pH, water-extractable total dissolved solids (TDSs), coexisting microbes, and the sampling site. The results of this study will help improve our knowledge of *E. coli* survival in sediments and determine the sediment factors that should be considered in numerical models for their survival.

## 2. Materials and Methods

### 2.1. Sediment Sampling

For laboratory experiments, sediment samples were collected from three lakes in Yamagata Prefecture, Japan, on 9 December 2023: Lakes A (38.7643 N, 139.7594 E), B (38.72584 N, 139.9052 E), and C (38.6724 N, 139.9015 E). The sediment samples were collected using an Ekman barge sampler at least three times from the center of each lake within a radius of approximately 1–2 m and well mixed to obtain a stable sample. The mixed samples were passed through a 2 mm sieve and stored in a cooler box using sterile 2 L polyethylene bottles. Then, the sieved samples were transported to the laboratory and kept overnight at 4 °C in the dark. Subsequently, a subsample of each precipitated sediment sample was autoclaved at 121 °C for 1 h for sterilization, and each of the sterilized and untreated samples was stored at 4 °C in the dark until further use.

Lakes A, B, and C, which were used as irrigation water sources, had a water volume of 30,000.0 m^3^, 16,200 m^3^, and 50,000.0 m^3^, respectively. The sediment texture type was sandy loam in Lake A (9.4% clay, 38.3% silt, and 52.4% sand) and B (12.8% clay, 40.4% silt, and 46.8% sand), and loam in Lake C (7.5% clay, 29.6% silt, and 63.0% sand). Other general physicochemical properties (moisture content, organic matter content, and water-extractable dissolved organic carbon (DOC)) of the sediments are described in Appendix A.

Before the laboratory experiments, *E. coli* concentrations in the autoclaved and untreated sediment samples were measured as follows. Approximately 10 g-wet sample was diluted with 40 mL sterile saline and shaken for 5 min. The mixtures were stabilized for 5 min, and 100 μL of the upper sediment-free water samples were poured and spread on Chromocult Coliform Agar (Merck, Darmstadt, Germany). The agar plates were incubated at 37 °C for 24 h, and the blue colonies on the agar were counted as *E. coli*. This analysis was conducted in triplicate for the sediment sample collected in each lake, and colony counts were reported as the mean colony-forming units (CFUs) of the triplicate plates. As a result, *E. coli* concentrations in the untreated sediment at Lake A, B, and C were 8.9, <8.9, and 15.5 CFU/g-wet, respectively, while its concentration in the autoclaved sediments at all lakes was <8.9 CFU/g-wet (Appendix A).

### 2.2. Preparation of E. coli Strain

*E. coli* strain K-12 (NBRC 3301) was used for the laboratory experiment. This strain was obtained from the Biological Resource Center, National Institute of Technology and Evaluation (NBRC), Tokyo, Japan, and has been used in similar studies to evaluate the survival of this bacteria in various water environments [27,28,29,30]. Before the experiment, the strain was cultured at 37 °C for 24 h in 10 mL Luria–Bertani (LB) broth. To remove LB broth from the *E. coli* culture, the culture was centrifuged at 10,000× *g* for 5 min at 4 °C, and the pellet was resuspended and washed three times with 10 mL sterile saline [31]. This suspension containing *E. coli* at approximately 10^8^ CFU/mL was diluted 10-fold with sterile saline to achieve an *E. coli* concentration of approximately 10^7^ CFU/mL for laboratory experiments.

### 2.3. Laboratory Experiments

A series of laboratory experiments was conducted to observe the survival of *E. coli* in sediments by controlling any two sediment factors (temperature, pH, water-extractable TDSs, coexisting microbes, and sampling site), to determine which factor affected the survival of *E. coli* the most. Approximately 1–10 g of the sediment samples were dispensed into sterile 15 or 50 mL polypropylene tubes. The sediment samples with 2–3 cm height in the tubes were then inoculated with 10 µL *E. coli* suspension per 1 g-wet sediment, achieving its concentration of approximately 10^5^ CFU/g-wet. The inoculated samples in the tubes were mixed using a vortex mixer (Vortex-Genie 2, Scientific Industries, Inc., Bohemia, NY, USA) and incubated for 28 days without agitation under the following conditions: 20 °C, pH 6, no additional water-extractable TDSs, no coexisting microbes, and Lake A sediment. In addition, 78 treatments were examined in triplicate by controlling two of the above five factors under the conditions described in Table 1. Each factor was controlled according to the following procedure, described below.

#### 2.3.1. Temperature

During the experiments, the sediment samples in the tubes were stored at 10, 20, and 30 °C, which were determined based on typical lake sediment temperatures in Japan [32] and conditions in similar previous studies [13,14].

#### 2.3.2. pH

At the beginning of the experiments, the pH of the sediment samples was set to 4, 6, or 8, which is commonly observed in lakes worldwide [33]. To analyze the pH, the sediment was mixed in Milli-Q water at a 1:5 (g:mL) ratio, shaken for 30 min at 121 rpm, and measured using a portable pH meter (LAQUAtwin; HORIBA, Ltd., Kyoto, Japan). The pH of the sediment sample was adjusted to the aforementioned values using HCl and NaOH [34].

#### 2.3.3. Water-Extractable Total Dissolved Solids

Water-extractable TDSs were prepared by extracting dissolved solids from sediment collected from Lake A on 1 November 2023, using the abovementioned procedure. To extract TDSs, the sediment sample was first mixed with Milli-Q water at a ratio of 1:4 (g/mL) and shaken for 24 h. The mixture was then centrifuged at 10,000× *g* for 20 min and filtered through a polyvinylidene fluoride membrane filter of 0.65 μm (Merck, Darmstadt, Germany) and a polyethersulfone membrane filter of 0.22 μm (ASONE, Osaka, Japan) for particulate removal and sterilization [35]. The filtered water was stored at −80 °C until further analysis and freeze-dried using an FDU-1200 freeze dryer (EYELA, Tokyo, Japan).

Laboratory experiments were conducted with three additional water-extractable dissolved organic carbon (DOC) levels (0, 0.004, and 0.010 mg per 1 g-wet sediment), which were achieved by adding TDSs. The added water-extractable DOC volumes in these experiments were determined based on typical ranges observed in environmental lake sediments [36]. Water-extractable TDSs were dissolved in Milli-Q water and added to the sediment samples at a given DOC concentration, as measured using a TOC-L (SHIMADZU, Kyoto, Japan). The detailed composition of the water-extractable TDSs is listed in Appendix A.

#### 2.3.4. Coexisting Microbes

To assess the effects of coexisting microbes in lake sediments, including predation and competition, experiments were conducted using untreated and autoclaved sediments without predation and competition.

#### 2.3.5. Sampling Sites

Experiments were also conducted using sediment samples collected from the three lakes (Lake A–C) mentioned above to observe the effects of sediment sampling sites, such as sediment texture type and nutrient composition, on *E. coli* survival.

### 2.4. Data Analysis

#### 2.4.1. Determination of *E. coli* Population Change Rate

The dilution plate method was used to measure *E. coli* concentration in the tubes at 0, 1, 3, 7, 14, and 28 days [37]. The sediment samples in the tubes were diluted with sterile saline at a 1:4 (g:mL) ratio and shaken for 5 min. The mixtures were stabilized for 5 min, and the upper sediment-free water samples were diluted with sterile saline up to 10^−4^. In total, 100 μL of the diluted samples were poured and spread on Chromocult Coliform Agar (Merck, Darmstadt, Germany). The agar plates were incubated at 37 °C for 24 h, and the blue colonies on the agar were counted as *E. coli*. Colony counts were reported as the mean CFUs of triplicate plates.

The chick exponential model [38] is commonly used to simulate *E. coli* populations (Equation (1)).
*C* = *C*_0_*e^kt^*,(1)
where *C* is the *E. coli* concentration (CFU/g) at time *t*, *C*_0_ is the initial *E. coli* concentration (CFU/g), and *k* is the *E. coli* population change rate (d^−1^). Equation (1) can be transformed into a logarithmic form (Equation (2)).
ln *C =* ln *C*_0_ + *kt*,(2)
The linear least-squares method using *E. coli* concentrations was used to estimate the population change rate. However, later in the experiment, the concentration and time did not exhibit a linear relationship. In these cases, the data obtained from the beginning of the experiment, where a linear relationship was found, were used to estimate the population change rate.

#### 2.4.2. Relative Maximum *E. coli* Concentration

To evaluate the growth potential of *E. coli* in the sediment samples at different water-extractable TDS levels, the relative maximum *E. coli* concentration during the comparative experiments was calculated using the following equation:(3)$$\mathrm{Relative}\;\mathrm{maximum}\;\mathrm{concentration}=\frac{C_{\mathrm{Max}}}{C_{0}}$$
where *C*_Max_ is the maximum *E. coli* concentration (CFU/g).

#### 2.4.3. Statistical Analysis

A two-way analysis of variance (ANOVA) was performed on the population change rates obtained from the experiments with the two controlled factors to examine whether either factor significantly affected the population change rate. Furthermore, a one-way ANOVA was conducted to examine the effect of the addition of water-extractable TDSs on the relative maximum *E. coli* concentration. Tukey’s honest significant difference test was used for post-hoc comparisons in both ANOVAs. These analyses were performed at a statistical significance level of *p* < 0.05.

The independent effect of each factor on the population change rate was ranked by comparing its effect sizes (η^2^) obtained in two-way ANOVA with a statistical significance level (*p* < 0.005) adjusted by the Bonferroni method considering the multiple comparisons of 10 pairs of factors. To create this ranking, in pairs containing a factor with a significant effect size, the factor with a higher effect size was placed at a higher rank than the other factors. This ranking can also be determined by multiple regression analysis (MRA) of the population change rates of *E. coli* using the given values of the considered factors. However, MRA may be affected due to the limited experimental conditions (i.e., most of the data are at 20 °C). Thus, this study directly compared the effect sizes of the two factors to determine their ranking. Statistical software R (version 4.2.2) was used for all the analyses.

## 3. Results and Discussion

### 3.1. E. coli Survival under Various Sediment Conditions

Figure 1 and Figure 2 show the changes in *E. coli* concentrations and the rate of population change of this bacterium during comparative experiments (focusing on two factors at a time). Under all conditions with the autoclaved sediments, *E. coli* survived for 28 days at all temperatures and grew at 20 and 30 °C (Figure 1a,b,d). The population change rates under the conditions with the autoclaved sediment of Lake A at 20 (−1.16 to 1.21 day^−1^) and 30 °C (−1.27 to 1.21 day^−1^) were higher than those at 10 °C (−1.14 to −0.12 day^−1^). The *E. coli* growth at above approximately 20 °C is consistent with a previous study reporting that *E. coli* can grow in autoclaved sediment [39,40]. Autoclaving sediments appear to provide nutrients that support the growth of *E. coli* and eliminate competitive and predatory coexisting microbes [40]. This provision of nutrients by the autoclave was consistent with the fact that the concentration of water-extractable DOC, which is an indicator of nutrients, was higher in the autoclaved sediments than in the untreated sediments (Appendix A). Under all conditions with the autoclaved sediments, *E. coli* did not grow at 10 °C, which is consistent with a previous study that reported that it could not grow at a cooler temperature (4 °C) in autoclaved sediment [39]. These cells of *E. coli*, which could not grow, may have been in a viable but non-culturable (VBNC) state, as reported at a low temperatures (<10 °C) [41].

The effect of temperature was insignificant (−0.33 to −0.27 day^−1^) under the conditions with the untreated sediments. However, the population change rate at 30 °C (−0.33 day^−1^) was slightly lower than those at 10 and 20 °C (−0.29 and −0.27 day^−1^, respectively), indicating a trend of enhanced *E. coli* die-off with increasing temperature in consistency with previous studies [13,14]. This trend could be explained by the possibility that coexisting microbes that prey on and/or compete with *E. coli* in the untreated sediment may be more active at warmer temperatures [42,43]. For example, a previous study reported that *Daphnia magna* and *Brachionus calyciflorus*, metazooplanktons, could remove *E. coli* in a warm water environment (22 °C) more actively than those at 10 °C [42].

The population change rates were lowest at pH 4 (−1.32 to −0.64 day^−1^), followed by pH 8 (−0.39–0.88 day^−1^) and pH 6 (−0.36–1.21 day^−1^) (Figure 1a,e,f,g). Previous studies have reported that although *E. coli* can survive a wide range of pH values (3–10) [44], the optimal pH for its survival in an aquatic environment is between 5 and 7 [15,16], revealing that the lowest population change rates at pH 4 and the highest population change rates at pH 6 found in this study are reasonable. In aquatic environments, especially eutrophic lakes, an increase in pH due to algal growth and photosynthesis can promote *E. coli* inactivation [45]. However, the results of this study indicate that even in aquatic environments with higher pH, attention should be paid to the long-term survival and growth of *E. coli* in sediments.

Figure 3 shows the relative maximum *E. coli* concentration during the experiments, focusing on the water-extractable TDS level. The relative maximum concentration with additional TDSs (1–510 and 1–196 in the 0.004 and 0.01 mg additional DOC levels, respectively) was higher than or similar to that without additional TDSs (1–123). In addition, the additional TDSs tended to delay the day of maximum concentration (Figure 1b,e,h,i). These results suggest that additional water-extractable TDSs enhance *E. coli* growth and long-term survival, which is consistent with studies reporting that higher organic matter, including nutrients, contributes to *E. coli* growth and long-term survival [17,18]. These results suggest caution against the growth and long-term survival of *E. coli* in sediment of aquatic environments receiving water enriched with organic matter, such as domestic, agricultural, and livestock wastewaters. However, the population change rates under the conditions using the autoclaved sediment with additional TDSs (−1.32–0.68 and −1.26–0.57 day^−1^ in the 0.004 and 0.01 mg additional DOC levels, respectively) were lower than or similar to those without additional TDSs (−1.16–1.21 day^−1^) (Figure 1b,e,h,i). These decreases in the population change rate caused by the addition of TDSs appear inconsistent with the higher maximum concentrations shown in Figure 3. The nutrients released from the TDSs added to the sediment should have contributed to the higher maximum concentrations and the longer growth period from the beginning to the peak concentration. The lower population change rates, which were calculated for the entire growth period with the addition of TDSs, were likely attributed to the slow growth of *E. coli* observed in the latter half of the longer growth period.

The population change rates in almost all conditions with the untreated sediment (−0.64–0.88 day^−1^) were significantly lower or comparable to those with the autoclaved sediment (−1.16–1.21 day^−1^). Previous studies reported that, in sediment environments, competition and predation inhibit the survival of *E. coli* [20,21]. Although this study could not distinguish whether the coexisting microbes in the sediments competed or preyed on *E. coli*, they inhibited their survival, which is consistent with the findings of previous studies [20,21].

The population change rates at Lake B (−1.28–0.30 day^−1^) and C (−1.28–0.25 day^−1^) were insignificantly different in all comparative experiments, while in some conditions (i.e., 20 °C, 30 °C, pH 6, no additional TDSs, and autoclaved sediment), the rates at Lake A (−1.16–1.21 day^−1^) were significantly higher than those at the other lakes. In addition to the factors considered in this study, sediment texture and nutrients can affect *E. coli* survival. Previous studies have reported that higher fine particle (silt and clay) and organic matter contents enhance the survival of *E. coli* in sediments [14,17]. In this study, the silt, clay, and organic matter contents in the sediments were the highest in Lake B (43.5, 18.2, and 17.3%, respectively), followed by Lakes A (38.2, 12.6, and 13.2%, respectively) and C (32.5, 7.2, and 13.1%, respectively) (Appendix A), showing no relationship between the population change rates and these contents. The silt, clay, and organic matter contents in the sediments used in this study were generally higher than in Chandran et al. at 0.51–15.7, 9.40–17.5, and 1.84–4.96%, respectively [17] and than in Garzio-Hadzick et al. at 9.59–20.7, 4.45–19.5, and 1.35–5.14%, respectively [14]. This may be why we did not find the same trend as previous studies, where *E. coli* survival is enhanced in sediments containing higher fine particle and organic matter contents. Further study is required to determine the physicochemical properties that cause differences in population change rates among different sediments.

In this study, the effects of changes in the sediment factors on population change rates were evaluated by using *E. coli* strain K-12. However, previous studies reported that *E. coli* survival in the beach sand [20] and river sediment [39] environment depends on its strain and phylogenetic group. While phylogenetic groups A, B1, and D/E can survive longer in the sand environment than other groups [20], dependency of *E. coli* survival in lake sediment environments on its strains and phylogenetic group is still unknown, motivating further studies.

### 3.2. Quantification of Effects of Factors on E. coli Population Change Rate

Table 2 compares the impact of two factors on the *E. coli* population change rate based on the effect size (η^2^) obtained by two-way ANOVA.

#### 3.2.1. Independent Effects of Each Factor

Based on the data in Table 2, the overall ranking of the independent effects of each factor on the rate of population change was determined, as shown in Figure 4. This overall ranking revealed that pH (η^2^ = 59.5–89.0%) significantly affected the population change rates (*p* < 0.005) the most, followed by coexisting microbes, sampling sites, temperature, and TDSs. This result was supported by MLR analysis using the population change rate and each factor as dependent and independent variables, respectively (Appendix A). Previous studies reported that pH is the main factor affecting *E. coli* survival in river sediments [46] and soils [47,48]. Coexisting microbes (46.0–48.4%) had a more significant effect on population change rates than other factors, except for pH (9.0 and 59.5%, respectively) and water-extractable TDSs (1.7 and 0.4%, respectively). The sampling sites had a moderate effect on the population change rate among the five factors considered in this study. Although temperature and organic matter content have also been recognized to affect *E. coli* survival [14,17], surprisingly, this study showed that temperature (16.2–25.8%) and TDSs (0.4–20.9%) had a smaller effect on population change rates than the other factors. Most previous studies have developed numerical models to simulate *E. coli* survival in sediments in aquatic environments, ignoring the effects of sediment factors [24,25]. For more reliable simulations, it is recommended that numerical models be developed with preferential consideration to pH and coexisting microbes, which have a higher rank in the independent effects of the sediment factors determined in this study.

#### 3.2.2. Interactive Effects of Each Factor

As shown in Table 2, significant interactive effects on *E. coli* survival were observed in all combinations of the two factors, implying complex relationships between each factor in the sediments. This indicates that the effect of one factor on the survival of *E. coli* in the sediment can be enhanced or weakened by other factors. The effect size of the interaction between two factors was generally larger than that of at least one factor independently, except for the combination of temperature and pH (13.4%) and that between coexisting microbes and sampling site (22.3%). The interactions related to temperature, pH, and water-extractable TDSs were probably due to changes in temperature, pH, or TDS levels, which could affect the activity of *E. coli* and coexisting microbes [49,50]. However, the detailed mechanisms of each interaction remain unclear, necessitating further investigation.

The interaction with pH, which had the largest independent effect on *E. coli* survival, was most significant for coexisting microbes (31.1%), followed by temperature (13.4%), sampling site (9.9%), and water-extractable TDSs (7.4%). The interaction with coexisting microbes with the second largest independent effect on *E. coli* survival was most significant for water-extractable TDSs (85.4%), followed by pH (31.1%), temperature (26.4%), and sampling site (22.3%). Based on these results, we recommend that numerical models simulating *E. coli* survival consider not only pH and coexisting microbes as independent variables but also the interactions between pH and coexisting microbes, coexisting microbes, water-extractable TDSs, coexisting microbes, and temperature.

In this study, the effect size of each sediment factor on population change rates and their rankings were determined based on experiments under limited conditions (Table 1) simulating the lakes. Thus, the applicability of these results to other sediment environments should be examined experimentally using a range of conditions suitable for the target environment. On the contrary, because the survival of *E. coli* in the sediments is affected by a complex combination of factors, such as chemical composition, further investigations are needed to determine the effect of these factors on their survival.

## 4. Conclusions

In this study, to determine the independent and interactive effects of sediment factors on *E. coli* survival in lake sediments and to determine the factors that strongly affect their survival, comparative experiments were conducted by controlling for sediment factors, including temperature, pH, water-extractable TDSs, coexisting microbes, and sampling site. The pH was found to have the most significant effect on the survival of *E. coli*, followed by the presence of coexisting microbes, sampling site, temperature, and water-extractable TDSs. Significant interactions in *E. coli* survival were observed between two factors. Among the significant interactions related to pH and coexisting microbes that had large independent effects, relatively large interactions were observed between pH and coexisting microbes, coexisting microbes and water-extractable TDSs, and coexisting microbes and temperature. These results suggest the need to develop numerical models for simulating *E. coli* survival that consider both the independent and interactive effects of multiple factors, according to the priorities determined in this study.

The finding of this study could help future studies develop numerical models that can simulate *E. coli* survival in lake sediments. The methodology used in this study may be applied to obtain information required for such model developments in coastal and river sediments.

## Figures and Tables

**Figure 1 microorganisms-12-01192-f001:**
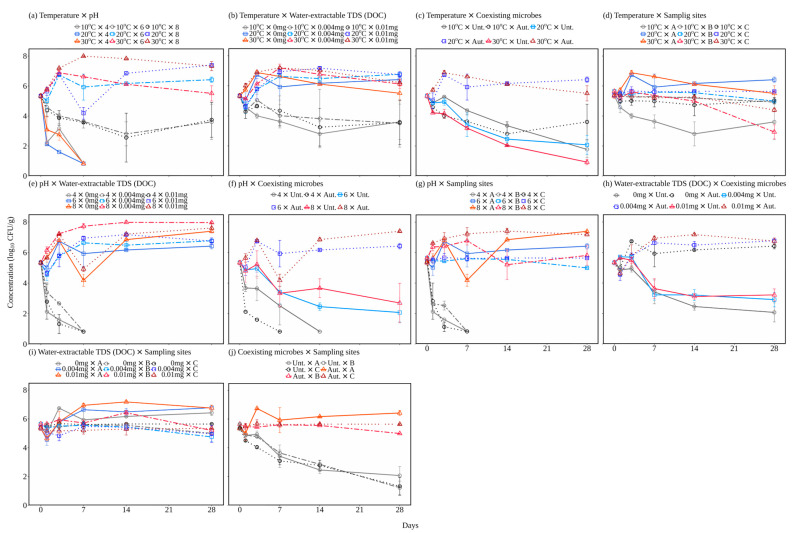
Changes in *E. coli* concentration in the laboratory experiments. The subtitles of each figure describe the two sediment factors being controlled in each experiment. Abbreviations: Unt., untreated sediment; Aut., autoclaved sediment (no protozoa or bacteria).

**Figure 2 microorganisms-12-01192-f002:**
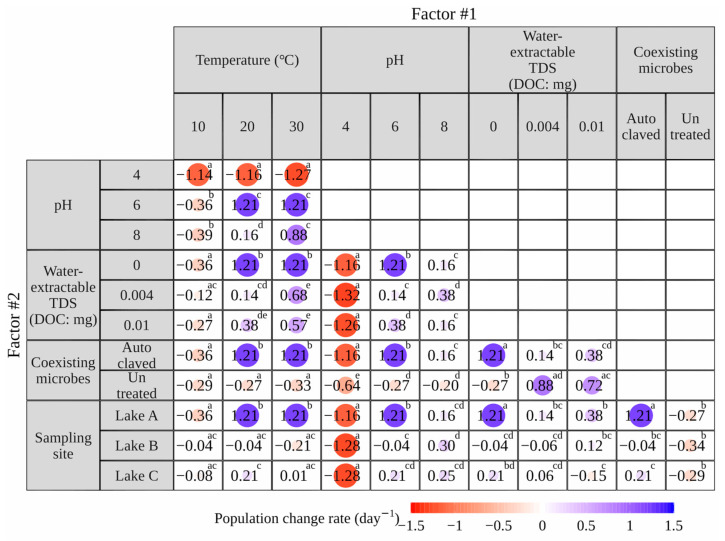
*E. coli* population change rate in the laboratory experiment focusing on two sediment factors. Positive and negative values represent growth and die-off, respectively. In each series experiment, different letters indicate significant differences (*p* < 0.05), whereas similar letters indicate no significant difference.

**Figure 3 microorganisms-12-01192-f003:**
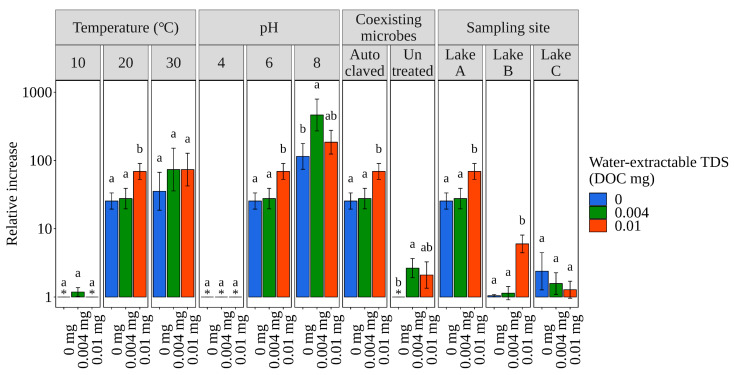
Relative maximum *E. coli* concentration during the comparative experiments focusing on the conditions of water-extractable TDS addition. In each treatment, different letters indicate significant differences (*p* < 0.05), whereas similar letters indicate no significant difference. An asterisk indicates that the relative increase was less than 1.

**Figure 4 microorganisms-12-01192-f004:**
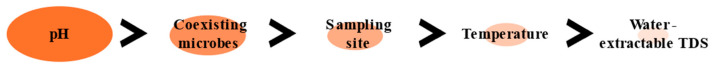
Order of the effect size of each factor on *E. coli* population change rate as the summary of comparisons of two parameters in Table 2.

**Table 1 microorganisms-12-01192-t001:** Levels of each sediment factor set in laboratory experiments.

Parameter	Levels		
Temperature (°C)	10	20	30
pH	4	6	8
Added water-extractable TDSs (concentration of DOC: mg/g-wet)	0	0.004	0.01
Coexisting microbes	Untreated sediment	Autoclaved sediment (no protozoa and bacteria)	
Sampling site	Lake A	Lake B	Lake C

**Table 2 microorganisms-12-01192-t002:** Comparison of the effects of two factors on *E. coli* survival based on the effect size (η^2^: %) obtained in two-way ANOVA.

Factors (#1 × #2)	Effect Size (η^2^: %)		
	Factor #1	Factor #2	Interaction #1 × #2
Temperature × pH	16.2 *	68.9 *	13.4 *
Temperature × Water-extractable TDSs	65.9 *	14.4 *	16.5 *
Temperature × Coexisting microbes	25.8 *	46.0 *	26.4 *
Temperature × Sampling site	22.5 *	36.1 *	36.9 *
pH × Water-extractable TDSs	89.0 *	3.2 *	7.4 *
pH × Coexisting microbes	59.5 *	9.0 *	31.1 *
pH × Sampling site	84.1 *	4.8 *	9.9 *
Water-extractable TDSs × Coexisting microbes	0.4	1.7	85.4 *
Water-extractable TDSs × Sampling site	20.9 *	44.3 *	30.2 *
Coexisting microbes × Sampling site	48.4 *	26.6 *	22.3 *

* *p* < 0.005 (a statistical significance level adjusted by the Bonferroni method).

## Data Availability

The original contributions presented in the study are included in the article/Appendix A, further inquiries can be directed to the corresponding author.

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
