# Peer review of "Significant Factors for Modelling Survival of Escherichia coli in Lake Sediments"

_microorganisms, 2024, doi:10.3390/microorganisms12061192_

Round 1

Reviewer 1 Report

Comments and Suggestions for Authors

The authors in the study propose several new combinations of factors for the numerical modeling of E.coli development in water bodies. This issue is of high practical importance whereas the conclusions are supported by the results. Nevertheless there a re few serious points that need to be addressed

In the abstract some more specific results are needed. For instance, how much was the pH effect? Was it statistically significant? Please provide some numbers to support the results 

The introduction is well written and presents the previous literature, however lacks a part that introduces the reader concerning the experiments performed, which I recommend to be added. 

Materials and Methods 

The samplings are from different lakes? It is not clear, please write it clearer in the first paragraph of the introduction. If this is the case, why not from different points of the same lake to keep some factors stable? Please provide a short explanation

How did the authors examine the presence of E.coli in samples before the preparation of the strains and the laboratory experiments? If E.coli was already in high populations in the sediments this would mislead the results. Or how was this factor addressed?

Results and discussion part lacks reference support in some cases. The temperature related results should be more extensively discussed and supported by references. One would expect significant change. I suggest to discuss more extensively this part

Author Response

Response to comments from Reviewer #1

General Comment: The authors in the study propose several new combinations of factors for the numerical modeling of E.coli development in water bodies. This issue is of high practical importance whereas the conclusions are supported by the results. Nevertheless there a re few serious points that need to be addressed.

Response: We appreciate the reviewer’s insightful comment as well as your interest in our research. Thank you very much for your constructive suggestions, which have considerably enhanced the quality of the manuscript. As noted below, the manuscript has been revised.

  • Specific Comment #1: Abstract: In the abstract some more specific results are needed. For instance, how much was the pH effect? Was it statistically significant? Please provide some numbers to support the results.

    Response:
    Thank you for bringing this to our attention. pH (effect size: η2 = 59.5–89.0%) had statistically larger effect on E. coli population change rates than other factors. In addition, the effect size of coexisting microbes (46.0–48.4%) was statistically higher than that of other factors, except for pH and water-extractable TDS, and was higher (not statistically) than that of water-extractable TDS (1.7 and 0.4%, respectively). We have added these effect sizes based on statistical analysis into Abstract of the revised manuscript.

  • Specific Comment #2: Introduction: The introduction is well written and presents the previous literature, however lacks a part that introduces the reader concerning the experiments performed, which I recommend to be added.

    Response: Thank you for drawing our attention to this. We have added introductions of the experiment conditions conducted in previous studies, which can be seen in Lines 46–53 of the revised manuscript.

  • Specific Comment #3: Materials and Methods: The samplings are from different lakes? It is not clear, please write it clearer in the first paragraph of the introduction. If this is the case, why not from different points of the same lake to keep some factors stable? Please provide a short explanation.

    Response: Thank you for drawing our attention to this. We collected sediment samples from three different lakes. We have revised it clearly in the materials and methods section, which can be seen in Lines 78–80 of the revised manuscript. In addition, although we did not collect sediment samples from different points of the same lake, we collected samples at least three times from the area, the center of each lake within a radius of approximately 1–2 m, and mixed them to keep some sediment factors stable. These additional explanations can be seen in Lines 80–83 of the revised manuscript.

  • Specific Comment #4: Methods: How did the authors examine the presence of E.coli in samples before the preparation of the strains and the laboratory experiments? If E.coli was already in high populations in the sediments this would mislead the results. Or how was this factor addressed?

    Response: We measured E. coli concentration in sediment samples before the laboratory experiments by using the dilution plate method with Chromocult Coliform Agar (Merck). Consequently, as shown in Table S1, the E. coli concentration in these samples (≤ 15.5 CFU/g wet) was clearly smaller than the initial E. coli concentration in the laboratory experiments (approximately 105 CFU/g wet). This means that the indigenous E. coli in these samples did not disturb the results of the laboratory experiments. These explanations and results can be seen in Lines 94–104 of the revised manuscript.

  • Specific Comment #5: Results and discussion: Results and discussion part lacks reference support in some cases. The temperature related results should be more extensively discussed and supported by references. One would expect significant change. I suggest to discuss more extensively this part.

    Response: Thank you for bringing this to our attention. Following the reviewer’s comment, we have added additional discussions about the results of this study, especially the results related to temperature, which can be seen in Results and discussions section of the revised manuscript (Lines 220–224, Lines 237–244, Lines 249–253, and Lines 287–288).

Reviewer 2 Report

Comments and Suggestions for Authors

The authors aimed to study the independent and interactive effects of sediment factors on E. coli survival in lake sediments to develop numerical models for simulating E. coli survival. Results indicated that pH had the strongest independent effect on E. coli survival, followed by the presence of coexisting microbes, sampling site, and temperature.

The overall subject of the manuscript has scientific relevance, and fits the Microorganisms topics. 

Please see some suggestions for improvement below:

1. To reinforce the importance of the study consider add information in the introduction about:

a) E. coli in sediment is not taken into account when performing water quality evaluation of the water bodies.

b) Diarrheagenic E. coli (DEC) can cause disease.

c) Highlight that sediment can accumulate bacteria, and thus can serve as reservoir. Bacteria, such E. coli can form biofilms and persist in the environment. In this biofilm the bacteria have a privilege environment to share virulence and antibiotic resistance genes.

d) The WHO list of antibiotic-resistant “priority pathogen” where E. coli is recognized as priority level 1 (critical).

2. Please improve the quality of the pictures, particularly Figure.

3. Line 54: Consider change to ”(e.g., high organic carbon content and small particle size), and their disturbance”.

4. Line 64: Consider to add “to feed the prediction models”.

5. Line 66; “The effects of each factor on E. coli survival were statistically significant.” Independent, interactive, or both effects? Please clarify.

6. Line 77: Consider to change “portion” to “subsample”.

7. Line 98: How do you control the inoculum size?

8. Please provide information on the arrangement of the sediment in the tubes/slurries. What is the height of the sediment in the tubes. Did you use any agitation?

9. Line 109: Consider change add “research conditions”.

10. Line 138: Remove subsection heading.

11. Line 186: Change heading to “Results and Discussion”.

12. Line 189: Change to “focusing on two factors at a time”.

Comments on the Quality of English Language

The authors showed a good command of the English language and overall, the manuscript flow is good and the information is clear. Minor editing of English language required. 

Author Response

Response to comments from Reviewer #2

General Comment: The authors aimed to study the independent and interactive effects of sediment factors on E. coli survival in lake sediments to develop numerical models for simulating E. coli survival. Results indicated that pH had the strongest independent effect on E. coli survival, followed by the presence of coexisting microbes, sampling site, and temperature.

The overall subject of the manuscript has scientific relevance, and fits the Microorganisms topics.

Response: We are extremely appreciative of the reviewer's comments, which have considerably enhanced the quality of the manuscript. As noted below, the manuscript has been revised.

  • Specific Comment #1: Introduction: To reinforce the importance of the study consider add information in the introduction about:
    a) E. coli in sediment is not taken into account when performing water quality evaluation of the water bodies.
    b) Diarrheagenic E. coli (DEC) can cause disease.
    c) Highlight that sediment can accumulate bacteria, and thus can serve as reservoir. Bacteria, such E. coli can form biofilms and persist in the environment. In this biofilm the bacteria have a privilege environment to share virulence and antibiotic resistance genes.
    d) The WHO list of antibiotic-resistant “priority pathogen” where E. coli is recognized as priority level 1 (critical).
    Response: We agree with the reviewer’s comments. Thus, in the revised manuscript, we have provided the following information in Lines 41–52, 69, and 73–74:
    1) E. coli in sediments is not considered in water quality evaluation of the water bodies,
    2) diarrheagenic and antibiotic-resistant coli are present, and
    3) E. coli can survive in sediments associated with or by forming biofilms.

  • Specific Comment #2: Figures: Please improve the quality of the pictures, particularly Figure.

    Response: Thank you very much for drawing our attention to this. The resolution of all figures has been improved in the revised manuscript.

  • Specific Comment #3: Introduction Line 54: Consider change to ”(e.g., high organic carbon content and small particle size), and their disturbance”.

    Response: Thank you for bringing this to our attention. Following the reviewer’s comment, the word has been replaced in the revised manuscript (Line 71).

  • Specific Comment #4: Introduction: Line 64: Consider to add “to feed the prediction models”.

    Response: Our study also aimed to provide the information for developing numerical models that simulate E. coli survival. Thus, we agree with the reviewer’s comment and have added this phrase to the revised manuscript (Lines 82–83).

  • Specific Comment #5: Introduction: Line 66; “The effects of each factor on E. coli survival were statistically significant.” Independent, interactive, or both effects? Please clarify.

    Response: In almost all experimental conditions except for a condition in which the level of water-extractable TDS and the presence of coexisting microbes were controlled, both the independent effects of each factor and the interaction effect of the two factors on E. coli survival were statistically significant. This result was described in Subsection 3.2. of the revised manuscript. However, since it is not appropriate to show the specific results in the introduction section, we decided to remove this sentence from the paragraph (Lines 80–88).

  • Specific Comment #6: Materials and Methods: Line 77: Consider to change “portion” to “subsample”.

    Response: We agree with the reviewer’s comment, and the word has been replaced in the revised manuscript (Line 98).

  • Specific Comment #7: Materials and Methods: Line 98: How do you control the inoculum size?

    Response: Before the laboratory experiment, we analysed the concentration of E. coli in the suspension after cultivating and washing it with sterile saline, following the procedure shown in the manuscript. As a result, the E. coli concentration in this suspension was approximately 108 CFU/mL. Thus, the suspension was diluted 10-fold with sterile saline to achieve its concentration of approximately 107 CFU/ mL, and 10 µL of the diluted suspension was added into 1 g wet sediment to achieve the target concentration (105 colony-forming units (CFU)/g wet). This additional explanation of E. coli inoculum size can be seen in Lines 123–127 and 133–135 of the revised manuscript.

  • Specific Comment #8: Materials and Methods: Please provide information on the arrangement of the sediment in the tubes/slurries. What is the height of the sediment in the tubes. Did you use any agitation?

    Response: We used sterile 15 or 50 mL polypropylene tubes, and the sediment height in the tubes was 2–3 cm. In addition, the sediment sample in the tubes was mixed using a vortex mixer after E. coli inoculation. However, during the incubation for 28 days, the samples were stored without any agitation. We have provided this additional information on the arrangement of the sediment, which can be seen in Lines 133–137 of the revised manuscript.

  • Specific Comment #9: Materials and Methods: Line 109: Consider change add “research conditions”.

    Response: Thank you for bringing this to our attention. Following the reviewer’s comment, we have added the phrase “conditions in similar previous studies” to the revised manuscript (Line 154).

  • Specific Comment #10: Materials and Methods: Line 138: Remove subsection heading.

    Response: We appreciate the reviewer bringing our attention to this subsection heading. Since the subsection heading was incorrect, it has been changed from “Coexisting microbes” to “Sampling sites” in the revised manuscript (Line 183).

  • Specific Comment #11: Results and Discussion: Line 186: Change heading to “Results and Discussion”.

    Response: Following the reviewer’s comment, the word has been changed in the revised manuscript (Line 232).

  • Specific Comment #12: Results and Discussion: Line 189: Change to “focusing on two factors at a time”.

    Response: Following the reviewer’s comment, this sentence has been replaced in the revised manuscript (Lines 235–236).

Reviewer 3 Report

Comments and Suggestions for Authors

The authors present an assessment of the factors that contribute to E. coli survival in sediments, which can be an important reservoir for this organism. The data are presented clearly, and the figures make it easy to understand. Figure 2 was especially effective at conveying the factors most important to survival. I do think though, there should be some discussion as to the relevance of collecting these data for E. coli K12. this is known to have many mutations that may not represent the state of E. coli out 'in the wild'. It is also well known the diversity of E. coli, and that some phylogroups appear to have better capabilities of surviving in extra-host environments. It would be useful to point these things out in the discussion.

Author Response

Response to comments from Reviewer #3

General Comment: The authors present an assessment of the factors that contribute to E. coli survival in sediments, which can be an important reservoir for this organism. The data are presented clearly, and the figures make it easy to understand. Figure 2 was especially effective at conveying the factors most important to survival. I do think though, there should be some discussion as to the relevance of collecting these data for E. coli K12. this is known to have many mutations that may not represent the state of E. coli out 'in the wild'. It is also well known the diversity of E. coli, and that some phylogroups appear to have better capabilities of surviving in extra-host environments. It would be useful to point these things out in the discussion.

Response: We are extremely appreciative of the reviewer's comments, which significantly improved the quality of the manuscript.
We selected E. coli K12 as the fecal indicator bacteria because this strain was frequently used in similar previous studies to evaluate the survival of this bacterium in various water environments (DeVilbiss et al., 2021; Gao et al., 2024; Suzuki et al., 2019; Wang et al., 2022). We have added this explanation, which can be seen in Lines 106–109 of the revised manuscript. However, we agree with the reviewer’s comment that differences in strain and phylogroups in E. coli may affect the factors that influence the survival of this bacterium. We have provided this additional discussion, which can be seen in Lines 321–327 of the revised manuscript.

Round 2

Reviewer 1 Report

Comments and Suggestions for Authors

The authors addressed successfully my comments, I suggest publication

Author Response

Response to comments from Reviewer #1

General Comment: The authors addressed successfully my comments, I suggest publication.

Response: We appreciate the reviewer's critical comments prior to the publication of our manuscript.